# Extension of TAM by Perceived Interactivity to Understand Usage Behaviors on ACG Social Media Sites

**Jui-Hsiang Lee** [1,2,]* and **Chang-Franw Lee** [1]

[1] Graduate School of Design, National Yunlin University of Science and Technology, Yunlin 64002, Taiwan; leecf@yuntech.edu.tw

[2] Department of Digital Multimedia Design, China University of Technology, Taipei City 116, Taiwan

[*] Correspondence: leock@gm.cute.edu.tw; Tel.: +886-2-29313416-2831

**Abstract:** Social media sites, as participatory artistic online platforms, have become popular spaces for amateur artists to exchange the creative artefacts of animation, comics and games (ACG) with others. Online social media platforms for ACG amateur artists offer interactive features for communication, information exchange and distance learning to facilitate connections among amateur artists worldwide. However, not too much of the available research has regarded investigating the determinative factors of ACG users' behavior on social media. By integrating aspects of perceived interactivity and users' willingness to exchange information, this study formulated a concept model to investigate the prerequisite factors of users' "continuance use intention" on ACG social media sites. The snowballing method was used to recruit 367 participants who had experience in creating ACG works and using more than two ACG social media sites from the Japanese sub-cultural communities at a university in Northern Taiwan. The results of this study provided empirical evidence supporting perceived interactivity as a prerequisite factor for extending the technology acceptance model (TAM), perceiving interactivity, and supporting indicators such as continuance use intentions and the willingness to exchange information on ACG social media sites.

**Keywords:** ACG; social media; perceived interactivity; continuance use intention; willingness to exchange information

## 1. Introduction

Social media, as participatory artistic online platforms, have become popular spaces for amateur artists to exchange animation, comics and games (ACG) creative artefacts with others. Social media encourage users to share, communicate and cooperate with others, and enable users to actively connect with others. Their great capabilities have gained increasing popularity, fueled by the demands of ACG enthusiasts to collect and share creative content and distribute related knowledge. These platforms have become popular sites for amateur artists and fandoms, and have engaged individuals with original design and functions in user-created artwork delivery and distribution, ranging from art and comics sites (e.g., DeviantArt and Pixiv) to video sites (e.g., Niconico (previously Nico Nico Douga) and Bilibili (the B site)) and also photo sites (e.g., Twitter and Instagram).

DeviantArt (dA), launched in 2000, is an online social platform for artists and art appreciators. According to dA's current openings for staff recruitment page of the site has over 26 million members and 251 million submissions [1]. DeviantArt demonstrates their members' artefacts for every fan to see, comment on and consume; the number of visitors in one day immensely exceeds that of any renowned museum [2]. Based on public voices, Salah [3] argued that dA has replaced a role of the *Salon des*

*Refuses*, and has thus become an enormous global artefact market, presenting a new exhibition type for users' evaluation and consumption.

In addition to dA, Pixiv (Shibuya, Tokyo, Japan), founded in 2007, is also a crucial social media site in the world for amateur artists and fandoms. Obtaining an official declaration from Pixiv on 28 January 2012, it is seen that the number of users had broken 4 million, monthly views totaled 2.8 billion, and the number of works published on Pixiv reached almost 24.5 million [4], and up until 22 February 2014, the number of users by then had been more than 10 million [5]. This shows how the influence of dA and Pixiv as ACG creative online artistic exchange platforms has manifested itself worldwide.

ACG amateur artists need to collect and distribute creative content and share information to help them cope in interactions with other users. With the current trend of cooperative interaction and co-creation, it is noted that the Internet plays a crucial role in facilitating dialog between amateur artists and their fans [6]. The comic industry has even devised a way (e.g., involving their fans to partner with them in their product creation process) to actively interact with ACG social media sites [6,7].

Like any other profit-seeking industry, the ACG industry further developed with the evolution of media and the advancement of information technology (IT) [8,9]. For example, in the past, traditional editors of comic magazines gave their opinions on the main decision-making directions throughout the whole comic magazine publishing process, and were responsible for making major decisions with respect to content creation. At present, however, editors of a comic magazine welcome their readers to be a partner to help rate the manuscripts being published, and seek new, talented artists using conventions and contests. They also started to take part in social media sites, sometimes on Pixiv, actively searching for interactions with amateur artists and fans. They are totally different from traditional editors [6,7]. Comic social media users can express their views after paying attention to specific content, uploading illustrations, providing comments and sharing content uploaded by other users. These platforms not only provide media users with a sense of belonging to a community, but also enable them to control and maintain their content and manage content permissions [10], which greatly increase the exchange activities of people with similar interests around the world.

Watabe and Abe [7] mention the importance of the dōjinshi culture in the Japanese comic industry, and they argue that social media sites such as Pixiv create alternative opportunities for the practice of Japanese comic dōjinshi culture and the fandom of the English-speaking world, and that Pixiv users are organized in circles of similar interests and share their artefacts. They obtained the point of Jenkins, Ito & Boyd [11], that the participatory culture of the Pixiv site was not created by the site itself, but by the people who use the site. They note that the collaborative culture was not developed by Pixiv or other similar social media, but is particularly originated in dōjinshi culture. Dōjinshi is a combination of the three stanzas, "dō" which means "the same", "jin', meaning "person", and "shi", meaning "magazine". Therefore, the "Dōjinshi" can be accepted as a magazine which includes artbooks, light novels, ACG features and is published by an individual enthusiast that targets specific hobby fandoms and is passionate about something together [7,12]. Pixiv can be regarded as a part of the dōjin culture within Japanese comic history, which is the cause of users on Pixiv having common hobbies and sharing their knowledge and creations, similar to how simulated dōjin groups operated in circles [7].

The rapid development and widespread introduction of new communication technologies is a challenge to individuals of new media, as well as to the study of Interactivity. While broader fandom studies are beginning to examine the concrete ways whereby online technologies can support "transcultural" communication [13,14], fannish community-building and practices of activities in reality, few studies have focused upon how fans of ACG use online technologies to interact.

Without such empirical research, it is difficult to judge how effective these 'multidirectional communications technologies of the Internet' [13] actually are at helping ACG fans of various national, linguistic and cultural backgrounds establish connections, and what kind of interactions these technologies encourage and realize.

Smart users use social media to get their advantages, whenever they exchange user-created content (UCC) or user-generated content (UGC) with others. Even though the UCC of social media sites is so popular, what makes users participate in these UCC sites? The answer is still not clear. Kane, Alavi, Labianca and Borgatti [15] pointed out that users in a social media environment seem to care more about the performances between their relationships, and interactions neglect the purpose of their communication. Thus we review the TAM and existing technology continuance use literature to identify the driving forces of technology usage.

By using TAM perspectives, Kim, Karatepe, Lee and Demiral [16] found the perceived enjoyment construct has a positive impact on the attitude toward using social media among females. The perceived usefulness was a positive influence on the attitude toward using social media among males, and thus are more seemly to appear at festivals. Sullivan and Koh [17] extend a dual-factor model of technology used to measure 268 Facebook users, and found that the prime enabler factor of social media continuance intention is the perceived enjoyment, while the prime inhibitor is the perceived complexity. These findings refer to the idea that there might have prerequisite factors that influence the users' affection for perceived enjoyment or perceived complexity. Abdullah, Kamal, Azmi, Lahap, Bahari and Din [18] try to formulate a research model by developing from the TAM to explore the perceived interactivity construct, and expect to impact the mediator of perceived usefulness for understanding the hotel online booking intention.

Interactivity, from a technical point of view, is the fundamental capability of interpersonal communication practices or reciprocal messages among senders, receivers and their interfaces, which suggests that web interactivity design can facilitate web-based communication [19,20]. Online social media is a popular emerging phenomenon. If the perceived interactivity of ACG social media and its influence on continuance use intention can be well understood, then we can apply it to web interactivity design or social media sites of different themes, in order to improve user interaction and interpersonal communication on the Internet. The adoption of new technologies or the popularity of predictability will help the related stakeholders facilitate a discussion of academic theories and an application of implementation practices, and more effectively improve the efficient use of resources.

We aimed to study the determinants of users' behavior on social media, and whether perceived interactivity can improve social media interactivity design and extend the technology acceptance model (TAM) to the ACG social media environment. Based on this purpose, the following three questions were investigated:

(1) Is the perceived interactivity of users supported as a prerequisite variable in the TAM on ACG social media sites?
(2) After users' perceived interactivity serves as the prerequisite variable in the TAM on ACG social media sites, are other variables still supported in the TAM?
(3) Is the willingness of users to exchange information supported as a dependent variable on ACG social media sites?

## 2. Literature Review

### 2.1. ACG Social Media Professional Techniques

Kaplan and Haenlein [21] defined social media as a 'group of Internet-based applications that build on the ideological and technological foundations of Web 2.0, and that allow the creation and exchange of user-generated content'. Thus, they embody user-generated content (UGC) and Web 2.0 these two concepts. Web 2.0 refers to platforms on which all users can change the contents and applications in active, cooperative ways, rather than limiting this activity to particular users. UGC refers to the various kinds of contents on the media available to the public and generated by registered individuals cooperatively [19,21].

Boyd and Ellison [22] argued that social media services are web-based services which permit an individual to (1) build a public or semi-public profile within a boundary system, (2) clearly declaim a

list of partners with whom they share a connection, and (3) view and transit their list of connections, as well as the connection lists of others within the system.

The primary aim of social media services is to build communities of participants and encourage interactions among them. The possibility of exchanging and discussing works created by artists and fans makes ACG social media services extra attractive to fans. The last decade has seen a proliferation of social media services explicitly designed to help users publish and discuss their own media, instead of merely interacting with other users. Therefore, designers and managers still need to determine how to make a social media service into an ideal platform for important artist and fan activities.

Chen [23] interviewed dōjinshi artists of anime conventions, who are described as anime/manga fans and active cultural producers. They are engaged in the reproduction of the materials they consume and in the manipulation of ideas, meanings and cultural references they perceive. Chen also listed four incentives to describe why they join the ACG fan culture: (1) Enjoying the experience of being a successful artist; (2) enjoying free imagination and fantasy through drawing or cosplay (costume play); (3) fulfilling the urge to perform, and one's sense of curiosity; (4) participating for social reasons.

Therefore, the current ACG participatory online artistic space has the following functions: (1) Submission, search and focus on artistic works; (2) a mechanism for ranking, discovery, and recommendation; (3) derivation of common interests or creation of accomplices through favorite tags and tag editing; (4) build groups; (5) sponsorship and sales; (6) official contests and activities planned by users. For example, dA combines several interactive facilities to provide not only an art-related online space, but also a community dedicated to sharing the user-generated artwork of amateur artists and users, as well as art and artwork divulgation, which enhances the appearance of opportunities to present artwork. One of the promotion mechanisms in dA is the way of choosing around 25 or more artworks every day and publishing them on the homepage of dA for a one-day duration [3]. Five tips were suggested to get your artwork noticed on dA: (1) Regularly submitting a certain number of works of good quality; (2) learning from teaching materials to improve your skills; (3) presenting yourself professionally; (4) getting to know or making friends and fans in the forum; (5) frequently browsing artworks and commenting [24].

Pixiv user evaluations are intended to be immediately passed onto the artist, so that users can identify the extent to which others' artwork is seen or the chance of it being seen by others. This is a major attraction for Pixiv users. Like many other online media, these evaluations are reflected in Pixiv's ranking system, which is more or less a common feature with other sites [6,7]. Furthermore, Pixiv allows people to freely connect with any other entity they wish, sometimes publicly, and establish a role profile or persona [25]. Watabe and Abe [7] assumed that Pixiv implements the tagging function only to facilitate searching for other users' artwork, but members have redefined the tags as a means of communicating in different ways, partly because Pixiv allows its members to freely create their own tags. More precisely, users who use the tagging feature can not only build a brand with their favorite artwork, but also search for their own requests. Pixiv users use these tags to expand their participation in the online media [7]. Pixiv also allows users to build groups and events and to manage events. Pixiv FANBOX is a communication platform for creators and fans, allowing users to enjoy a unique way of communication brought about by closed spaces. Here, fans can support creators and constantly interact with them through long-term sponsorship. From these monthly sponsorship fees provided by fans, creators can obtain the funds they need for creative activities on a regular basis. BOOTH is a work complex for creators.

Pixiv and dA both offer their registered users interactive features, and enable the online space to share not only artwork but also ideas, techniques, tutorials, reusable images, etc. In addition to the option to register to evaluate other members' artwork, one can leave messages, write comments, organize campaigns and activities and create groups on specific themes. Pixiv also allows its users to add and view records and give bookmarks, ratings, comments, tags and 'applause' to others [7].

In terms of the orientation of human-to-information interaction, when you submit and upload work, the social media site has already set the submission options for you (e.g., title and description

of the submission, image size, number of sheets and whether you are willing to tag at the same time). Then, recommended tags or category tags pop up for you to choose from. Subsequently, there are options, such as whether you wish to set any browsing restrictions, or are willing to submit to Twitter. After submission, you can also manage your work using functions such as sorting by page view and access analysis. In terms of human-to-human interaction, the interface design of these information systems is based on the artworks, which makes the interaction between humans more frequent and smoother.

## 2.2. Technology Acceptance Model

The Theory of Reasoned Action (TRA) can be used to study the rational choices of individuals. TRA [26] is mainly used to analyze how an individual's attitudes intentionally impact the individual's behavior. On the one hand, the individual's behavior may be reasonably inferred from the behavioral intention. On the other, an individual's behavioral intention may be predicted based on his or her attitude towards behavior and subjective norms. This gives people a clear understanding of the rational production of behavior. The theory has an important implicit assumption: People have the ability to fully control their behavior. However, in an organizational environment, individual behavior is subject to management intervention and external environment constraints. Therefore, it is necessary to introduce some external variables, such as situational variables, to meet the needs of research.

TAM is a model for studying the acceptance of information systems based on TRA [27]. In 1989, Davis used TAM to explain the general determinants of the computer acceptance of usage behavior. TAM proposes two notable factors—perceived usefulness and perceived ease of use—both of which are primary drivers of technology acceptance. The belief of an individual in a system may be influenced by other factors referred to as external variables in TAM [28]. The final version of TAM was formed by Davis and Venkatesh [29] after both perceived usefulness and perceived ease of use were found to have a direct effect upon behavior intention, so the need for the attitude construct was excluded.

But even though perceived usefulness and perceived ease of use both have been verified as salient predictors of IT acceptance, the following researches have shown that the fact of perceived ease of use tends to wear out over time on continuance use intentions, as users become increasingly familiar and acclimatized with IT usage [30,31]. One possible explanation is that perceived ease of use is probably not a relevant construct for IT continuance use intentions, just as Bhattacherjee and Barfar mentioned [32]. Another reasonable comprehension is that a potential individual using a new IT for the first time is required to overcome significant learning barriers; hence, IT that is viewed as being substantially difficult to use is less likely to be accepted by wary users, because such IT has already been filtered out at the beginning of its use.

It can be seen from the above discussion that despite the controversies surrounding any perceived ease of use and perceived usefulness in different studies, a perspective of why ACG social media is adopted as a creative online artistic exchange platform was not studied in prior research. Therefore, there is still a need for empirical research to validate various variables of TAM.

## 2.3. Actual and Perceived Interactivity

Interactivity in computer-mediated environments (CMEs) is regarded as a key scientific and technological capability that any individual can easily apply to access large amounts of online information and to show their quality [33]. Most of the existing research has examined the interactivity of websites [34] or web marketing [35], while little attention has been paid to IT, such as social media sites, especially ACG information usage behavior.

To some extent, interactivity epitomizes the promise of media technologies to capture the essence of interpersonal communication. Wu [36] argued that actual interactivity can be focused on a technical feature of the media, the capability of creating two-way information [37], or the potential for interactive communication in general [38]. Perceived interactivity is understood as a basic requirement in the message exchange process, regardless of whether the interaction occurs in face-to-face, wireless,

or non-face-to-face situations [39]. Hence, perceived interactivity appears to play a crucial role in understanding the effects of actual interactivity on interpersonal connection.

Yoo, Lee and Park [40] argued that perceived interactivity impacts utilitarian value. Perceived interactivity is also related to customers' online trust. Customers have greater intentions of engaging with media content when they trust it. Jeon, Jang and Barrett [41] found that perceived interactivity influences the intention of repurchase via perceived utilitarian value and online trust. Lin and Chang [42] found that users can utilize the interactive feature in Facebook to find health information to satisfy their outcome expectations and self-management competence. Mollen and Wilson [43] think that perceived interactivity can be categorized in a structuralized or mechanistic approach, and in the case of bad design, it has an adverse effect on consumer attitudes to websites. Some other empirical studies position interactivity as an antecedent of another experiential construct to support its placement in the conceptual framework.

Perceived interactivity is a construct with multiple dimensions. There is no consensus on the definition of interactivity. Previous literature has argued that the use of an information communication system requires individuals to be willing to transfer their resources to others affected by the interactive facilities [42,44]. However, another viewpoint, proposed by Rafaeli [38] (p. 117), is that actual interactivity can only provide the capabilities to facilitate the occurrence of interaction, because interactivity is potential adequacy, but is realized by communicators [38] (p. 117). Four perspectives of perceived interactivity can be categorized on the basis of the primary focus in the literature: Perceived interactivity as a characteristic of technology, perceived interactivity as a process of information exchange, perceived interactivity as a user perception, or combined approaches [45]. Liu and Shrum [46] defined interactivity as the mutual influences between two or more communication parties, including communication media and messages, as well as the synchronization magnitude of such influences (p. 54). To facilitate the conceptualization of interactivity, Sundar et al. [47] defined interactivity as a process in which communication parties including users, media and messages must be interchangeable for full interactivity (pp. 34, 35).

The above definitions of interactivity both involve users, media and messages. Although Interactivity has different definitions, a consensus theme is that the social media site successfully provides information to the individual, is perceived as responsive, and allows a sense of connection— often with other individuals. Therefore, in social media, perceived interactivity can be generally divided into two different dimensions: Human-to-human interaction and human-to-information interaction [42,48–50].

Lin and Chang [42] investigated the effects of such motivations on health information exchange on social media sites. They defined human-to-human interaction as the use of social media sites to send and receive information and obtain the reciprocal responsiveness of others, and human-to-information interaction as the use of social media sites to acquire and share information.

Their results demonstrate that the four following: Human-to-human interaction, human-to-information interaction, outcome expectations for health self-management competence and outcome expectations for social relationships, have a significant impact on health information exchange behavior.

In the context of current research, perceived interactivity results in users' positive reactions to websites and beliefs about them, such as efficiency, effectiveness, utilization, hedonics, enjoyment, or trust, which in turn influences behavioral intention, such as e-loyalty. This is consistent with TAM, in which the intention to accept or use a new technology is determined by perceived usefulness and perceived ease of use of the technology [51]. Following the above-mentioned studies of Sundar et al. [47] and Qiao [19], we attempted to advance the TAM by employing the interactivity of users perceived as a prerequisite factor for facilitating this study in a meaningful manner. Therefore, we proposed the following hypotheses:

**Hypothesis 1a.** *Human-to-human interaction has a positive effect on users' perceived ease of use of ACG social media sites.*

**Hypothesis 1b.** *Human-to-human interaction has a positive effect on users' perceived usefulness of ACG social media sites.*

**Hypothesis 2a.** *Human-to-information interaction has a positive effect on users' perceived ease of use of ACG social media sites.*

**Hypothesis 2b.** *Human-to-information interaction has a positive effect on users' perceived usefulness of ACG social media sites.*

## 2.4. Continuance Use Intention

We also observed that theories have been employed to study IT continuance use intention, such as TAM [27,28]. To understand the underlying factors that impact technology continuance use, IT continuance use behavior has been examined in research with theories, such as the extended expectation-confirmation model [52], theory of planned behavior [53], unified theory of acceptance and use of technology [54] and the cognition-affection-conation chain [55].

More generally, IT continuance use intention refers to IT post-adoptive behaviors. In Bhattacherjee's view, it refers to individual users' long-term use of an IT after its initial acceptance [31]. Drawing on the service retention literature of marketing, the expectation-confirmation theory (ECT) posits that users' IT continuance use intention is based upon their prior IT usage experience and their expectation of future benefits from continued IT usage. Prior research employed continuance use intention as the primary dependent variable [32]. Since the actual behavior is difficult to measure, it is quite common to measure the behavioral intention as a surrogate to the actual behavior, because intention is proven to be a valid predictor of the actual behavior.

In the IT acceptance model, perceived usefulness and perceived ease of use are considered to be the most prominent factors shaping user intention [28]. Perceived usefulness, also called performance expectation, is defined as the degree to which individuals' perceptions on using a certain IT would improve their performance of relevant tasks; researchers declared it can have positive influences on behavioral intention, too [56]. Perceived ease of use, also called effort expectation, is defined as the degree to which users believe that learning how to use an IT or actually using an IT would be free from efforts, or fewer efforts in comparison to other alternatives [57].

As previously discussed and obtained from Bhattacherjee and Barfar [32], they argued that subsequent researches have shown that the fact of perceived ease of use on continuance use intentions tends to wear out over time, because users become increasingly familiar with IT usage [30,31]. Therefore, IT that is considered difficult to use is unlikely to be accepted by discreet users. Another possible reason is that there are too many social media sites at present, resulting in a variety of user choices and too high a substitution rate. Users can choose to use the IT that is easy to use and give up the IT that is difficult to use. When IT is seen as a utility, it triggers a structure of perceived usefulness with the goal of improving user productivity and performance in the workplace.

Given the evolving nature of IT and the many benefits it can offer users (in addition to the benefits of productivity), perceived usefulness may become unimportant after a while. Although perceived ease of use plays a less important part in continued use behavior determination [58], researches on IT systems continuance use intention have discussed recently that the influence of the perceived ease of use on continuance use intention would be more crucial than perceived usefulness [59,60].

The above discussion insists that perceived ease of use and perceived usefulness have been controversial in different studies. However, in previous studies, no researchers expressed any views on why ACG social media sites would be adopted as participatory online artistic spaces. Therefore, it is considered necessary to conduct empirical research to verify the TAM. Hence, we proposed the following hypotheses:

**Hypothesis 1c.** *Human-to-human interaction has a positive effect on users' continuance use intentions towards ACG social media sites.*

**Hypothesis 2c.** *Human-to-information interaction has a positive effect on users' continuance use intentions towards ACG social media sites.*

**Hypothesis 3.** *Perceived ease of use has a positive effect on users' continuance use intentions towards ACG social media sites.*

**Hypothesis 4.** *Perceived usefulness has a positive effect on users' continuance use intentions towards ACG social media sites.*

*2.5. Willingness to Exchange Information*

While intention is often a reasonable predictor of behavior, and especially so in continuance contexts, where users are already using the target IT, Bhattacherjee and Barfar [32] noted that intention is not the equivalent of behavior; there may be cases in which individuals intend to take action in a certain way, but the actual action is very different from their intention. Hence, we followed their recommendation that future research operationalize and measure IT usage behavior, rather than stopping at intention.

Since the introduction of the Internet, the speed and means of sharing information has changed greatly. Amateur artists in ACG participatory online artistic spaces demand to distribute and collect information and share creative content, and their demand for innovative thinking inevitably increases, which helps them to cope with other users and to create value [7].

Many researchers agree that perceived interactivity is a basic factor that motivates users to participate in social media to seek and share information. Yan et al. [61] report that reputation, self-worth and social support are users' main concerns when they share health information in an online health community. Johnston et al. [62] consider that information utility and social support are the two major benefits derived from online health community participation. The willingness to exchange information is crucial for publishing industries to improve their performance and achieve competitive advantage. Therefore, users' willingness to exchange information is an important act in continuing to use online artistic space social media sites.

The willingness to exchange information is defined as the openness and intention to share information between social exchange partners and also determines the extent to which information is shared [57]. In this study, we added the process by which users decide what information can be shared and came up with the following hypothesis:

**Hypothesis 5.** *Continuance use intention has a positive effect on users' willingness to exchange information on ACG social media sites.*

## 3. Research Method

*3.1. Research Framework*

Based on the previous discussion, we formulated five major hypotheses to propose a conceptual research model (Figure 1) and tested the hypotheses in the social media environment. This model aims to identify the antecedents of users' behavior by proposing a research model based on the technology acceptance model (TAM) theory. Perceived interactivity is involved as a prerequisite construct that together affects people's perceived ease of use and perceived usefulness, which in turn drives their continuance use intentions to explore how it affects users' willingness to exchange information.

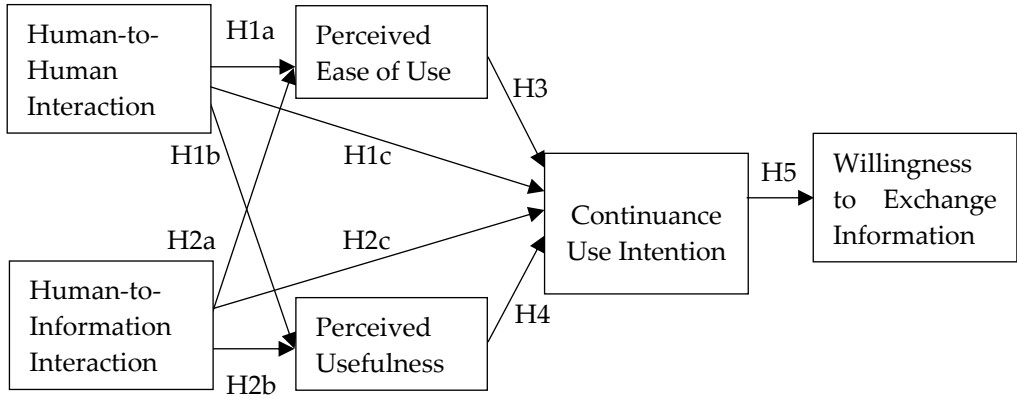

**Figure 1.** The concept model of research.

### 3.2. Sample Description

Students have the opportunity to communicate over the Internet and engage in a computer-mediated communication environment, and are thus a representative group for analysis. Walczuch and Lundgren [63] asserted that students were acceptable participants for electronic retail research. The main participants in previous animation, comics and games (ACG)-related activities (such as animation conventions) were students. Therefore, we recruited students as the main body of the sample.

We analyzed the possible responses of experienced users of ACG participatory online artistic spaces. This is because the ACG social media sites, especially the DeviantArt and Pixiv, did not put any unofficial surveys online. To overcome the challenges of ambiguous population size and samples recruit, purposive sampling was adopted over random sampling to recruit students mainly from Japanese sub-cultural communities, such as the anime community and the dōjinshi manga community at a university in Northern Taiwan. The snowballing method was utilized to extend the online questionnaire survey, and it is helpful to find anyone who has related experiences on ACG activities. Our model applies when a user's behavior is voluntary, and when the user has experience in more than two ACG social media sites, they were invited to join the questionnaire investigation.

A total of 383 participants were recruited for the questionnaire survey. During the response period of the questionnaire, 367 questionnaires were valid after the 16 questionnaires with missing or doubtful answers were excluded. There were 168 male participants and 199 female participants. In terms of usage habits, close to 52.5% of participants used ACG social media sites more than once per day. Close to 72.4% of participants used ACG social media sites for more than one hour. Samples are detailed in Table 1.

**Table 1.** Samples descriptions.

| **Gender** | **Male** | | | **Female** | |
|---|---|---|---|---|---|
| | **168** | | | **199** | |
| Frequencies of surfing on ACG websites (per day) | 1 | 2 | 3 | 4 | 5 |
| | 47.5 | 23.6 | 18.7 | 7.8 | 2.4 |
| Time spent on surfing ACG websites (per day) | 1HR below | | 1–2 HRS | More than 2HRS | |
| | 27.6 | | 39.5 | 32.9 | |

### 3.3. Scale Utilized and Data Collection

To ensure the validity of the scale, the variables in the study were operationalized using existing validated scales. Additionally, some items were slightly adapted to reflect the ACG social media context. Table 2 provides the survey items, items mean and the standard deviation (SD), along with their respective constructs and relevant references. Perceived interactivity can be divided into two

perspectives: Human-to-human interaction and human-to-information interaction. Human-to-human and human-to-information interactions were measured using the items adapted from Lin and Chang [42]. Perceived ease of use and perceived usefulness were measured using the items adapted and fixed from Davis et al. [58], Roca et al. [59] and Daruri et al. [64]. Continuance use intention was measured using the items adapted from Daruri et al. [64], Agarwal and Karahanna [65], Bhattacherjee and Premkumar [66], Thong et al. [60] and Wang, Xu and Chan [67]. Willingness to exchange information was measured using the items adapted from Lin and Chang [42], and we revised them to meet the ACG social media context. The questionnaire in the first section was answered based on a 7-point Likert scale, ranging from 1 (strongly disagree) to 7 (strongly agree). The items mean and SD can help the researcher outline the distribution of the users' replies.

**Table 2.** Respective constructs, survey items, means and relevant references. Note that ACG refers to animation, comics and games.

| Constructs /Items | Mean | SD | References |
|---|---|---|---|
| **Perceived Interactivity/Human-to-human interaction (PI-HHI)** | | | |
| PI-HHI1 I easily communicate with others using ACG social media sites. | 4.49 | 1.731 | [42] |
| PI-HHI2 I easily exchange opinions with others using ACG social media sites. | 4.73 | 1.830 | |
| PI-HHI3 I easily connect with other people using ACG social media sites. | 4.93 | 1.789 | |
| PI-HHI4 I easily develop interpersonal relationships with others using ACG social media sites. | 4.87 | 1.655 | |
| **Perceived Interactivity/Human-to-information interaction (PI-HII)** | | | |
| PI-HII1 I effectively submit artwork using ACG social media sites. | 5.49 | 1.272 | [42] |
| PI-HII2 I effectively filter information using ACG social media sites. | 5.43 | 1.314 | |
| PI-HII3 I effectively access interesting information using ACG social media sites. | 5.23 | 1.630 | |
| PI-HII4 I effectively obtain meaningful information using ACG social media sites. | 5.08 | 1.464 | |
| **Perceived ease of use (PEOU)** | | | |
| PEOU1 It is easy to surf ACG social media sites on my device. | 5.21 | 1.124 | [58,59,64] |
| PEOU2 It is easy to exchange ACG information on subject topics through ACG social media sites. | 5.06 | 1.421 | |
| PEOU3 I am easily given tags to ACG information by other users. | 5.43 | 1.436 | |
| PEOU4 The submitting on ACG social media sites is smooth for the device I use. | 5.21 | 1.132 | |
| **Perceived usefulness (PU)** | | | |
| PU1 The variety of materials on ACG social media sites covers all topics of interest for me. | 4.36 | 1.657 | [59,64] |
| PU2 The ACG social media sites are useful for me to exchange information. | 4.59 | 1.742 | |
| PU3 The ACG social media sites provide a comprehensive perspective on the topics of interest for me. | 4.61 | 1.735 | |
| PU4 ACG social media have sufficient functions for promoting my subject topics of creation. | 4.57 | 1.616 | |
| **Continuance use intention (CUI)** | | | |
| CUI1 I always like to try to use ACG social media sites. | 5.26 | 1.460 | [60,61,65–67] |
| CUI2 I keep using ACG social media sites as regularly as I do now. | 5.09 | 1.444 | |
| CUI3 I intend to continue using ACG social media sites over the next six months. | 5.25 | 1.425 | |
| CUI4 I will discontinue my use of ACG social media sites. | 4.92 | 1.625 | |
| CUI5 I will share and recommend others using ACG social media sites in the future as usual. | 5.04 | 1.601 | |
| UI6 I intend to spend more time on ACG social media as possible as I can. | 5.19 | 1.399 | |
| **Willingness to exchange information (WEI)** | | | |
| WEI1 I use social media sites for acquiring ACG-related information. | 5.31 | 1.323 | [42] |
| WEI2 I use social media sites for sharing ACG-related information. | 5.36 | 1.310 | |
| WEI3 I use social media sites for submitting ACG-related works. | 5.25 | 1.309 | |
| WEI4 I use ACG social media sites for finding people who have experienced problems similar to mine. | 5.04 | 1.487 | |
| WEI5 I use social media sites to corporate ACG-related works with others. | 5.36 | 1.511 | |

### 3.4. Data Analysis and Results

The Analysis of Moment Structures (AMOS) 21.0 and the statistical package for IBM SPSS 21.0 (Armonk, NY, USA) were used for the statistical analyses. AMOS was used because it supports covariance-based structural equation modeling (SEM) techniques and the objectives of this study. Convergent validity was measured with two metrics: Average variance extracted (AVE) and composite reliability (CR). All of the convergent validity metrics should exceed a minimum standard of 0.5 [68]. The reliability of the measurements was examined using CR. In general, the minimum acceptable value of CR is 0.7 [69]. For the current confirmatory factor analysis (CFA) model, the results showed that the CR of the constructs ranged from 0.868 to 0.9477. The AVE ranged from 0.570 to 0.695 (Table 3). Both of the two conditions for convergent validity were met.

**Table 3.** Convergent validity and reliability of measures in the research model.

| Construct | Convergence Validity | Reliability |
|---|---|---|
| | AVE | Composite Reliability |
| Human-to-information interaction (HII) | 0.622 | 0.868 |
| Human-to-human interaction (HHI) | 0.695 | 0.899 |
| Perceived usefulness (PU) | 0.653 | 0.881 |
| Perceived ease of use (PEOU) | 0.570 | 0.887 |
| Continuance use intention (CUI) | 0.750 | 0.947 |
| Willingness to exchange information | 0.597 | 0.872 |

The discriminant validity was assessed using the square root of the AVE and latent variable correlations (Table 4). The square root of the AVE of each construct should exceed the correlation shared among constructs, as this implies that constructs have good discriminant validity. In summary, the test results of the measurement model, including its convergent and discriminant validity measures, were satisfactory.

**Table 4.** Discriminant validity of the research model.

| Construct | HII | HHI | PU | PEOU | CUI | WEI |
|---|---|---|---|---|---|---|
| Human-to-information interaction (HII) | 0.637 | | | | | |
| Human-to-human interaction (HHI) | 0.000 | 1.732 | | | | |
| Perceived usefulness (PU) | 0.260 | 1.190 | 1.521 | | | |
| Perceived ease of use (PEOU) | 0.443 | 0.232 | 0.340 | 1.687 | | |
| Continuance use intention (CUI) | 0.492 | 0.282 | 0.399 | 0.353 | 0.766 | |
| Willingness to exchange information (WEI) | 0.374 | 0.214 | 0.303 | 0.268 | 0.581 | 0.905 |

The SEM approach was adopted in our data analysis, as it possesses many advantages over traditional methods. For example, the multivariate statistics technique integrating factor analysis and path analysis have multiple regression properties. This hierarchical causality is more in line with the human thinking form, which is not possible with traditional regression analysis [70]. A model is considered to have good model-data fit if the $\chi^2$ to degrees of freedom is smaller than 3, GFI and AGFI are above 0.80 and RMSEA is smaller than 0.08.

Table 5 illustrates the data after measurement of the proposed model. It can be seen that the $\chi^2$ to degrees of freedom was 2.679, CFI was 0.876 (<0.9), GFI was 0.861 (>0.8), AGFI was 0.833 (>0.8), and RMSEA was 0.067 (<0.08). The statistical analyses indicated that the proposed model had an acceptable goodness of fit (GOF).

**Table 5.** Indicators of model goodness-of-fit.

| Indicators | Value | Threshold | References |
|---|---|---|---|
| $\chi^2$ | 843.900 | | |
| Df | 315 | | |
| $\chi^2$/df | 2.679 | 1–3 (good), 1–5 (accept) | Schumacker and Lomax [71] |
| CFI | 0.876 | >0.9 (good) | Hu and Bentler [72] |
| GFI | 0.861 | >0.8 (accept) | Doll, Xia, and Torkzadeh [73] |
| AGFI | 0.833 | >0.8 (accept) | MacCallum and Hong [74] |
| RMSEA | 0.067 | <0.08 | Hu and Bentler [72] |

### 3.5. Hypothesis Testing Results

A structural model assessment was performed to examine the hypothesized relationships among the constructs. To elucidate the effects of perceived interactivity on user perceptions and thus to provide a comprehensive and meaningful conclusion, we analyzed the relationship between the hypotheses concerning perceived interactivity and its constructs. The path coefficients are shown in Table 6. Figure 2 displays the standardized path coefficients and path significances. All of the hypothesized relationships were significant with a p-value of less than 0.05, except for the relationships between perceived ease of use and continuance use intention and between perceived usefulness and continuance use intention.

As expected, Human-to-human interaction had a positive effect on perceived ease of use ($p = 0.013* < 0.05$), as well as on perceived usefulness ($p = *** < 0.001$). Therefore, H1a and H1b were supported. Human-to-information interaction had a positive effect both on perceived ease of use ($p = *** < 0.001$) and perceived usefulness ($p = *** < 0.001$), indicating that H2a and H2b were supported. Human-to-human interaction had a positive effect on continuance use intention ($p = 0.010 * < 0.05$) and Human-to-information interaction also had a positive effect on continuance use intention ($p = *** < 0.001$). Therefore, H1c and H2c were supported. However, perceived ease of use exerted an insignificant effect on continuance use intention ($p = 0.612 > 0.05$), and so did perceived usefulness ($p = 0.910 > 0.05$). Therefore, H3 and H4 were not supported. Continuance use intention had a positive effect on willingness to exchange information ($p = *** < 0.001$), indicating that H5 was supported.

**Table 6.** Estimation coefficients of the research model.

| Hypothesis | | Path | | SD | USD | CR | *p* | Y/N |
|---|---|---|---|---|---|---|---|---|
| H1a | HHI | –> | PEOU | 0.135 | 0.134 | 2.490 | 0.013 * | Yes |
| H1b | HHI | –> | PU | 0.733 | 0.687 | 11.804 | *** | Yes |
| H2a | HII | –> | PEOU | 0.427 | 0.695 | 6.438 | *** | Yes |
| H2b | HII | –> | PU | 0.264 | 0.408 | 5.096 | *** | Yes |
| H1c | HHI | –> | CUI | 0.241 | 0.160 | 2.569 | 0.010 * | Yes |
| H2c | HII | –> | CUI | 0.715 | 0.784 | 7.145 | *** | Yes |
| H3 | PEOU | –> | CUI | −0.030 | 0.068 | −0.507 | 0.612 | No |
| H4 | PU | –> | CUI | 0.011 | 0.008 | 0.113 | 0.910 | No |
| H5 | CUI | –> | WEI | 0.698 | 0.759 | 8.808 | *** | Yes |

* $p < 0.05$; ** $p < 0.01$; *** $p < 0.001$.

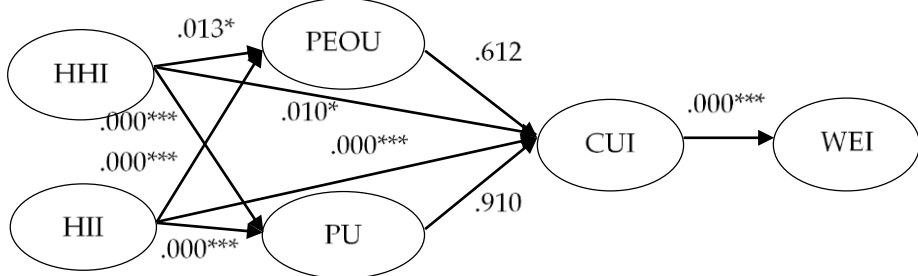

**Figure 2.** Results of the research model tests. * $p < 0.05$; ** $p < 0.01$; *** $p < 0.001$.

## 4. Results

The testing results (Figure 2) of H1a and H1b demonstrated that users' perceived ease of use and perceived usefulness were significantly affected by human-to-human interaction on ACG participatory social sites. The testing results of H2a and H2b showed that users' perceived ease of use and perceived usefulness were also significantly affected by human-to-information interaction on ACG participatory social sites. The testing results of H1c and H2c also demonstrated that continuance use intention was significantly impacted by human-to-human interaction and human-to-information interaction. Therefore, regarding the first question of this study, the statistical verification showed that users' perceived interactivity was supported as a prerequisite variable in the TAM on ACG social media sites.

Regarding the second question of this study, the statistical results of H3 and H4 showed that other variables were not supported in the TAM after users' perceived interactivity served as the prerequisite variable in the TAM on ACG social media sites. This is because users' continuance use intention was not affected by their perceived ease of use and perceived usefulness, but impacted by perceived interactivity more on ACG participatory social media sites. Since we add the perceived interactivity as the external factor to extend TAM, the effect of perceived ease of use and perceived usefulness seems comparatively not to be significant. This finding contrasts with the findings of some prior TAM literature, which is an interesting discovery of this study. We will discuss this in the next section.

According to the test results of H5, users' continuance use intention positively impacted their willingness to exchange information on ACG participatory social websites. Therefore, regarding the third question of this study, users' willingness to exchange information, which is a social media feature enabling users to create contents (mainly artwork exchange), was supported as a dependent variable on ACG social media sites.

The above statistical verification facilitates understanding of the factors that affect perceived interactivity on ACG social media sites and all TAM factors, as well as the important factors affecting users' willingness to exchange information on ACG social media sites.

## 5. Discussion

This study helps to facilitate understanding of the determinants of users' behavior on ACG social media sites. The statistical results of this study provide some implications for theories.

(1) The findings of this study supported H1a, H1b, H2a, H2b, H1c, H2c and H5. Perceived interactivity is involved as a prerequisite factor that affects people's perceived ease of use and perceived usefulness, which drives their continuance use intentions and then affects their willingness to exchange information. This was validated in this study. It can be seen that the interactivity design of a website is related to users' intentions towards activities on the website, which affects users' perceptions of interaction with the website and their intentions to continue using it.

(2) In the ACG social media context of the current study, perceived interactivity resulted in user reactions towards the website and beliefs about it, such as usefulness and perceived ease of use, and was related to an IT system leading to a continued behavioral intention and willingness to exchange information. In this study, the cognitive factors include perceived ease of use and perceived usefulness. When people have highly perceived interactivity on using ACG social media sites, they would perceive

the site easier and useful to use because they already have a high capability and ambition to use them. The effect of a specific IT system on perceived ease of use is also supported by the literatures. As Cyr et al. [51] insist that perceived interactivity is a strong predictor of an individual's behavior. The seemingly ambiguous effects of perceived interactivity on social media sites were clearly validated in this study.

(3) According to the statistical results of this study, H3 and H4 were not supported. The effects of users' perceived ease of use and perceived usefulness on their continuance use intentions towards ACG social media sites were not significantly supported. Lee, Kozar and Larsen [75] conducted a meta-analysis of 101 studies related to TAM published from 1989 to 2003. They discovered that 74 studies indicated a significant correlation among perceived usefulness, behavioral intention and actual behavior, while 58 studies indicated a significant correlation among perceived ease of use, behavioral intention and actual behavior. In other words, studies remain that do not support the correlation of perceived usefulness and perceived ease of use with other variables.

These findings may allow the viewpoint of Bhattacherjee and Barfar [32] to be reconfirmed. They argued that IT acceptance and continuance use intention are conceptually different behaviors, in that the former refers to users' first-time adoption of a new IT, while the latter refers to their long-term usage of an IT that is already in use. Given their fundamentally different nature, it can be reasonably speculated that the factors that predict continuance use intention may also be significantly different from those that predict IT acceptance. In addition, in the light of the evolving nature of IT and the multiple benefits it can offer to individuals (in addition to productivity benefits), we maybe can follow Bhattacherjee and Barfar [32] to employ expected benefits as predictors of continuance use intention, rather than perceived usefulness (which connotes productivity benefits only).

(4) Since Gu et al. [76] compared the employees and students use of Instant Messaging (IM). They found that employees consider utilitarian is priority factor while students are more influenced by perceived hedonic in their intention to use IM. So maybe the users' determinants of continuance use intention behavior have differences between different demographic groups. Glass and Li [77] found that social influence was more important in determining IM adoption than perceived usefulness (PU) and perceived ease of use. More than attitude or reasonable cognition, factors such as emotional or cultural reasons may be related, which needs to be further investigated. We can maybe follow Jenkins's viewpoint. Just as Watabe and Abe [7] think it is important, it needs to be noted that Pixiv itself does not determine its culture. People, not media, are the fundamental factors responsible for creating participatory culture [11]. Therefore, regarding website success, there seem to be other factors to further explore.

(5) As Watabe and Abe [7] highlighted, the tagging function allows members to freely create their own; they not only provide information to users in a contingent fashion, but also visually indicate the process of contingency based on their particular previous action on the interface. In the Human and Computer Interaction (HCI) domain, users use these tags to expand their participation in online social media. Scholars indicated the crucial component of displaying the "interaction history" of users with the IT system. Pelaprat and Shapiro [78] suggested to employ a "history of use" metaphor, for digital objects are not just an "ephemeral conversation". Instead, the history data display should be rich that could reflect and influence users' experience in many ways [20].

## 6. Conclusions

Innovative applications of interactivity on social media are crucial to attract and retain online users. Though there is the potential for interactivity provided by the social media site, little attention has been paid to how interactivity might be more utilized. The findings suggest that perceived interactivity has a positive impact on user-perceived ease of use and perceived usefulness, and has positive effects on user behavior which ultimately result in continuance use intention and willingness to exchange information. Hence, if online social media site designers and marketers wish to attract and stick users, then enhancement of web interactive features that enable users' perceived interactivity is desirable.

*6.1. Implementation Suggestions*

Since the rapid development of e-commerce online, organizations have rushed to launch their own branded websites. However, the number of visitors has gone from bad to worse. Because the one-way communication method of corporate websites is no longer attractive, it is replaced by social media websites, which can be freely published, discussed and freely interacted with visitors themselves. From a practical perspective, this study also has several implications suggestions for the management of social media sites.

(1) The results show that human-to-human interaction is an important factor that may increase users' perceived interactivity and willingness to exchange information on social media sites. Thus, to enhance the perception of human-to-human interaction, managers or designers of social media sites should provide some strategies to strengthen interaction among users. Management could encourage users to present their personal knowledge, such as achievements, exposure, discussion of works and topics and curation activities, etc. This would be to promote mutual attention and encourage comments sharing among users to increase interpersonal interaction. In addition, managers can enhance the compatibility of current popular technology teaching services and product promotion through various complementary services to increase human-to-human interactions as well.

Compared to social media platforms, a self-branding corporate site tends more to like one-way broadcasting, waiting for passive access, and may become less prominent. If the corporate sites can downsize to profiles, they may cooperate together to formulate a topic social media platform. Increasing more user-initiated activities and having users create/generate content to lead human and human interaction may greatly attract the users' attention.

(2) Human-to-information interaction is a predictor of information exchange. Although many artists and painters are involved in ACG information exchange sites, the tags interaction can improve interpersonal relationships and the tag settings of self-editing make the flexibility with emotions and are fun and popular for users. Huang et al. [79] find that emoticons, by means of text, may provide additional social cues beyond the electronic information to enhance the exchange of social information. Thus, management of social media sites could improve the richness of emoticons to strengthen visual cues and enhance user perceptions of virtual presence when browsing information on social media sites [42]. In addition, if the Pixiv users can submit their works to Twitter, why can it not synchronize to other theme-related social media sites as possible?

Another suggestion is the different themes of social media sites can be developed, and it can increase the discussion of topics and tag individual visitors to attract users to come together. Because the relationship between the retailer and consumer can be transformed into user and user is not often opposite, somehow these can become close partnerships through the complementarity of information exchanges.

(3) Additionally, Lin and Chang [42] argue that to enhance social interaction on social media, managers of social media may also embed some synchronous communication tools in information shared by users, which could allow users to conduct two-way conversations with an information contributor immediately. Recently, Pixiv has also incorporated channels for interaction among artists and sponsors, as well as for putting artwork in sales and on shelves. Although these belong to human-to-human design, prior categorization of information themes is still a good way to facilitate human-to-information interaction.

Therefore, we recommend that companies which envision converting social values of social media to commercial values should consider the impact of the orientations and characteristics of social media, and make a fit between marketing strategies and medium features. For example, Pixiv is more relationship-oriented than dA, possibly due to the influence of Japanese dōjinshi culture, so there are more derivative works on Pixiv. Users may care relatively more about communication cues which primarily convey interpersonal attitudes. Therefore, if managers apply this feature to different themes of user-generated websites, they can add interpersonal factors such as courtesy interaction to

commercial interactions and strike a balance between marketing tasks and interpersonal, emotional and cognitive concerns.

*6.2. Research Limitations and Future Study*

There are fewer limitations in this research. First, the questionnaire participants are limited to those students whose expanded samples were experienced on the ACG social media site. However, those individuals who have not experienced on the ACG social media site may become potential users. Therefore, further study may broaden the scope to recruit current and potential users to compare how this model applies to different populations. Second, the study respondents replied to the questionnaires after their using an ACG social media site based on the impression of their memories. It may not be precise according to their actual reaction on using an ACG social media site. Future studies could control the context variables in a laboratory and exam independent and dependent variables to make a better assessment. Finally, given the broad perspective of this study, there were possible variables not included in the research model to be verified. Future studies may explore the ACG social media usage behavior and attitude, such as hedonic or perceived in a community factor, from different approaches.

This ACG topic research is related to the issues of globalization and cultural diversity. There is hope that the exploration of the Extension of the TAM model by Perceived Interactivity to Understand Usage Behaviors on ACG Social Media Sites can help the ACG culture/industry, and that it has the scientific and integrated approaches to sustainable development in the future.

**Author Contributions:** Conceptualization, J.-H.L.; Methodology, J.-H.L. and C.-F.L.; Formal Analysis, J.-H.L.; Investigation, J.-H.L.; Data Curation, J.-H.L.; Writing-Original Draft Preparation, J.-H.L.; Writing-Review & Editing, J.-H.L. Visualization, C.-F.L.; Supervision, C.-F.L.

**Funding:** This research received no external funding.

**Acknowledgments:** Some of the materials in this article are presented as oral presentations entitled "Understanding The Continuance Use Intention On Social Media", 2019 Eurasian Conference on Biomedical Engineering, Healthcare and Sustainability. We especially thanks to the session chair and the participants for their constructive comments and given the Best Conference Paper Award in improving this paper.

**Conflicts of Interest:** The authors declare no conflict of interest.

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
