# Peer review of "Extension of TAM by Perceived Interactivity to Understand Usage Behaviors on ACG Social Media Sites"

_sustainability, doi:10.3390/su11205723_

Round 1

Reviewer 1 Report

Hello Authors

Minor Issues

Introduction section need review, while comparing with other sections introduction section need review Explain dōjinshi culture and importance: dōjinshi is used in these lines 66, 68 and 576 but not explained what is the culture and why it is important Line 531 needed reference. Line 57 to 65- ACG industry need more details or elaborate in the paragraphs TAM has recent articles with focus on social media, in the references TAM referred to old articles. You may refer classic articles. it is always advisable to give reference articles with in 5 years.

Author Response

Dear Reviewer

We sincerely appreciate your comments and suggestions regarding my manuscript “Extension of TAM by Perceived Interactivity to Understand Usage Behaviours on ACG Social Media Sites.” And for replying precise, we try to summarize your comments and transform them into 3 paragraphs. We believe we addressed all of your concerns in the revised manuscript. Our responses to your comments are listed as follows. Thank you so much for your time on this manuscript.

We also deliver a pdf file, please see the attachment.

Comment 1: Introduction section need review, while comparing with other sections introduction section need review Explain dōjinshi culture and importance: dōjinshi is used in these lines 66, 68 and 576 but not explained what is the culture and why it is important Line 531 needed reference.

Author’s response: We add two articles to compensate and refer to the definition of Dōjinshi and illustrate the importance of Dōjinshi culture.

    Watabe and Abe [7] mentions the importance of the dōjinshi culture in the Japanese comic industry and argues that social media sites such as Pixiv create alternative opportunities for the practice of Japanese comic dōjinshi culture and fandom of the English-speaking world and that Pixiv users are organised in circles of similar interests and share their artefacts. They obtained the point of Jenkins, Ito & boyd [11], the participatory culture of pixiv site was not created by the site itself but by the people who use the site. They note that the collaborative culture was not developed by pixiv or other similar social media, but particularly is originated in dōjinshi culture. Dōjinshi is a combination of the three stanzas, ‘dō’ which means 'the same', ‘jin’ meaning 'person', and ‘shi’ meaning 'magazine'. Therefore, the ‘Dōjinshi’ can be accepted as a magazine which includes artbooks, light novels, ACG features and published by an individual enthusiast that target specific hobby fandoms and passionate about something together [7,12]. Pixiv can be regarded as a part of the dōjin culture within Japan comic history, that's the cause of users on pixiv have common hobbies and sharing their knowledge and creations, similar to how dōjin groups operated in circles [7]. 

Please see Line 57 to 82.

Comment 2: Line 57 to 65- ACG industry need more details or elaborate in the paragraphs.

    Author’s response: We try to move the original line 69-74 to line 65-69 to compensate for the description of the ACG industry comic magazine editors’ elaborating.

    Like any other profit-seeking industry, the ACG industry further developed with the evolution of media and the advance­ment of information technology (IT) [8,9]. For example, in the past, traditional editors of comic magazines gave their opinions on the main decision-making directions throughout the whole comic magazine publishing process and were responsible for making major decisions with respect to content creation. At present, however, editors of a comic magazine welcome their readers to be a partner to help rate the manuscripts being published and seek new talented artists using conventions and contests. They also started to take part in social media sites sometimes on Pixiv for searching actively interactions with amateur artists and fans. They are totally different from traditional editors [6,7]. Comic social media users can express their views after paying attention to specific content, uploading illustrations, providing comments, and sharing content uploaded by other users. These platforms not only provide media users with a sense of belonging to a community but also enable them to control and maintain their content and manage content permissions [10], which greatly increase the exchange activities of people with similar interests around the world.   

Please see Line 65 to 69.

Comment 3: TAM has recent articles with focus on social media, in the references TAM referred to old articles. You may refer classic articles. It is always advisable to give reference articles within 5 years.

    Author’s response: We add four articles all within 5 years to describe the relationship between perceived interactivity and TAM constructs.

    Smart users use social media to get their advantages, whenever they exchange user-created content (UCC) or user-generated content (UGC) with others. Even though the UCC of social media sites is such popular, but what makes users participate in the UCC sites? still not unclear. Kane, Alavi, Labianca, & Borgatti [15] pointed out users in a social media environment seems more care about the performances between their relationships and interactions of neglect the purpose of their communication. Thus we review the TAM and existing technology continuance use literature to identify the driving forces of technology usage.

    By using TAM perspectives, Kim, Karatepe, Lee, Demiral [16] found the perceived enjoyment construct has a positive impact on attitude toward using social media among females. The perceived usefulness positive influence on attitude toward using social media among males and thus are more seemly to appear at festivals. Sullivan and Koh [17] extend a dual-factor model of technology used to measure 268 Facebook users and found that the prime enabler factor of social media continuance intention is the perceived enjoyment, the prime inhibitor is the perceived complexity. These findings refer there might have prerequisite factors that influence the users’ affection for perceived enjoyment or perceived complexity. Abdullah, Kamal, Azmi, Lahap, Bahari and Din (2019) try to formulate a research model by developing from the TAM to explore the perceived interactivity construct and expect to impact the mediator of perceived usefulness for understanding the hotel online booking intention.

Please see Line 92 to 109.

Reviewer 2 Report

Overall this is a strong article that draws on existing TAM research models and extends that work to ACG social media sites. The literature review lays a solid foundation for the methodology selected and the analysis is, for the most part, presented clearly. The findings mostly support prior research in this area, with the exception that the effect of perceived ease of use and perceived usefulness for continuance use intention did not rise to the level of significance. 

I have several suggestions for further clarifying the results and making the article even stronger. 

Some elaboration of dōjunshi culture in the introduction would be most helpful. This would help those who have a background in social media and interactivity research better connect to and frame the research being completed in this project.  Although minor, some discussion of the limitations of using a purposive recruitment method for participants should be included.  It may be helpful to provide the entire questionnaire used, even if it is inserted as an appendix. On first glance, Table 1 seems to reflect all of the questions used, but Table 2 suggests there were at least some additional demographic and use questions that were part of the survey. It would also be extremely helpful to include the average results of each question in the appendix, as this would allow for greater transparency in understanding the statistical analyses performed as well.  The conclusion section was by far the most problematic section of the paper. The first recommendation, running from line 537-552 includes a variety of strategies, but these do not seem to be backed up by the actual questions asked in the survey or by any citations. If suggestions such as these (e.g., including achievements) are going to be made, they must be supported by research or literature in some way. The references to hand-cranked and handshake tea (lines 546-552) were confusing. Although I have some knowledge of tea-drinking practices, these terms were new to me. Are hand-cranked and handshake teas different things? Perhaps more importantly, why is tea being used as an example in a paper focused on ACG? Is there a more relevant example? The language in this section also shifted to discussing "customers," from the point of view of the business, whereas the rest of the paper discussed the use of social media sites from the perspective of users. Why the shift? The second suggestion ends (lines 561-565) suggesting the development of themed social media sites. Is there evidence to suggest users are willing to engage with multiple themed social media sites? While some niche sites such as Deviant Art have had success, the most successful sites have been those of a general nature, such as Facebook, Twitter, Instagram, or Snapchat. Again, this suggestion did not seem to be backed by survey results or literature. Overall, the conclusion can be improved by more explicitly drawing links to the research that was completed as part of this project. This would also be a good place to explicitly link to concepts of sustainability, to make clear the relevance of this particular article to the journal. 

Author Response

Reviewer 2 to the authors:

Overall this is a strong article that draws on existing TAM research models and extends that work to ACG social media sites. The literature review lays a solid foundation for the methodology selected and the analysis is, for the most part, presented clearly. The findings mostly support prior research in this area, with the exception that the effect of perceived ease of use and perceived usefulness for continuance use intention did not rise to the level of significance.

I have several suggestions for further clarifying the results and making the article even stronger.

Some elaboration of dōjunshi culture in the introduction would be most helpful. This would help those who have a background in social media and interactivity research better connect to and frame the research being completed in this project. The conclusion section was by far the most problematic section of the paper. The first recommendation, running from line 537-552 includes a variety of strategies, but these do not seem to be backed up by the actual questions asked in the survey or by any citations. If suggestions such as these (e.g., including achievements) are going to be made, they must be supported by research or literature in some way. The references to hand-cranked and handshake tea (lines 546-552) were confusing. Although I have some knowledge of tea-drinking practices, these terms were new to me. Are hand-cranked and handshake teas different things? Perhaps more importantly, why is tea being used as an example in a paper focused on ACG? Is there a more relevant example? The language in this section also shifted to discussing "customers," from the point of view of the business, whereas the rest of the paper discussed the use of social media sites from the perspective of users. Why the shift? The second suggestion ends (lines 561-565) suggesting the development of themed social media sites. Is there evidence to suggest users are willing to engage with multiple themed social media sites? While some niche sites such as Deviant Art have had success, the most successful sites have been those of a general nature, such as Facebook, Twitter, Instagram, or Snapchat. Again, this suggestion did not seem to be backed by survey results or literature. Overall, the conclusion can be improved by more explicitly drawing links to the research that was completed as part of this project. This would also be a good place to explicitly link to concepts of sustainability, to make clear the relevance of this particular article to the journal.

Dear Reviewer

We sincerely appreciate your comments and suggestions regarding my manuscript “Extension of TAM by Perceived Interactivity to Understand Usage Behaviours on ACG Social Media Sites.” And for replying precise, we try to summarize your comments and transform them into 5 parts. We believe we addressed all of your concerns in the revised manuscript. Our responses to your comments are listed as follows. Thank you so much for your time on this manuscript.

We also prepare a pdf file, please see the attachment.

Comment 1: Some elaboration of dōjunshi culture in the introduction would be most helpful. This would help those who have a background in social media and interactivity research better connect to and frame the research being completed in this project.

Author’s response: We add two articles to compensate and refer to the definition of Dōjinshi and illustrate the importance of Dōjinshi culture.

    Watabe and Abe [7] mentions the importance of the dōjinshi culture in the Japanese comic industry and argues that social media sites such as Pixiv create alternative opportunities for the practice of Japanese comic dōjinshi culture and fandom of the English-speaking world and that Pixiv users are organised in circles of similar interests and share their artefacts. They obtained the point of Jenkins, Ito & boyd [11], the participatory culture of Pixiv site was not created by the site itself but by the people who use the site. They note that the collaborative culture was not developed by Pixiv or other similar social media, but particularly is originated in dōjinshi culture. Dōjinshi is a combination of the three stanzas, ‘dō’ which means 'the same', ‘jin’ meaning 'person', and ‘shi’ meaning 'magazine'. Therefore, the ‘Dōjinshi’ can be accepted as a magazine which includes artbooks, light novels, ACG features and published by an individual enthusiast that target specific hobby fandoms and passionate about something together [7,12]. Pixiv can be regarded as a part of the dōjin culture within Japan comic history, that's the cause of users on Pixiv have common hobbies and sharing their knowledge and creations, similar to how dōjin groups operated in circles [7]. 

Please see Line 70 to 82.

Comment 2: Although minor, some discussion of the limitations of using a purposive recruitment method for participants should be included. It may be helpful to provide the entire questionnaire used, even if it is inserted as an appendix. On first glance, Table 1 seems to reflect all of the questions used, but Table 2 suggests there were at least some additional demographic and use questions that were part of the survey. It would also be extremely helpful to include the average results of each question in the appendix, as this would allow for greater transparency in understanding the statistical analyses performed as well. 

    Author’s response: We analysed the responses of experienced users of ACG participatory online artistic spaces. Because the ACG social media sites, especially the DeviantArt and Pixiv, didn’t put any unofficial surveys online. To overcome the challenges of ambiguous population size and samples recruit, purposive sampling was adopted over random sampling to recruit students mainly from Japanese sub-cultural communities such as the anime community and the dōjinshi manga community at a university in Northern Taiwan. The snowballing method was utilized to extend the online questionnaire survey. And it’s helpful to find who have related experiences on ACG activities. Our model applies when a user’s behaviour is voluntary and when the user has experience in more than two ACG social media sites were invited to join the questionnaire investigation.

Please see Line 386 to 394.

Author’s response: And for transparency reason in understanding the statistical analyses suggested by the reviewer, and for the solid and concrete focus on our research topic we only put the average results of each question/item measured by whole respondents into Table 1. Consider the statistics results that must present after the reply of respondents. So we move the paragraph from Line 354-367 (origin) to Line 402-417. And Table 1 move to Table 2.

Please see Line 402-417.

Comment 3: The conclusion section was by far the most problematic section of the paper. The first recommendation, running from line 537-552 includes a variety of strategies, but these do not seem to be backed up by the actual questions asked in the survey or by any citations. If suggestions such as these (e.g., including achievements) are going to be made, they must be supported by research or literature in some way. The references to hand-cranked and handshake tea (lines 546-552) were confusing. Although I have some knowledge of tea-drinking practices, these terms were new to me. Are hand-cranked and handshake teas different things? Perhaps more importantly, why is tea being used as an example in a paper focused on ACG? Is there a more relevant example? The language in this section also shifted to discussing "customers," from the point of view of the business, whereas the rest of the paper discussed the use of social media sites from the perspective of users. Why the shift?

Author’s response 3-1: In this part, we may follow the upper paragraph. The authors want to insist on human-to-human interaction is an important factor that may increase users’ perceived interactivity and willingness to exchange information on social media sites. Thus, to enhance the perception of human-to-human interac­tion, managers or designers of social media sites should provide some strategies to strengthen interac­tion among users. Management could encourage users to present their personal knowledge such as achievements, exposure, discussion of works and topics, and curation activities, etc.

Please see Line 565-573.

Author’s response 3-2: And the authors just want to utilize an example to explain how the application runs. But maybe it made complexity. So we delete an example and revise the whole paragraph to make it clear.

Compare to social media platforms, a self-branding corporate site tends more like one-way broadcasting, waiting for passive access and may become less prominent. If the corporate sites can downsize to profiles, unite together to formulate a topic social media platform. Increasing more user-initiated activities and users create/generate content to lead human and human interaction maybe can greatly attract the users’ attention.

Please see Line 574-578.

Author’s response 3-3: if the Pixiv users can submit their works to Twitter also why it cannot synchronize to another them related social media sites as possible?

Another suggestion is the different themes of social media sites can be developed. And it can increase the discussion of topics and tag individual visitors to attract users to come together. Because the relationship between the retailer and consumer can be transformed into user and user is not often opposite, somehow it can become close partnerships through the complementarity of information exchanges.

Please see Line 188-196. Especially Line196.

Please see Line 586-592. Especially Line 589-592

Comment 4: The second suggestion ends (lines 561-565) suggesting the development of themed social media sites. Is there evidence to suggest users are willing to engage with multiple themed social media sites? While some niche sites such as Deviant Art have had success, the most successful sites have been those of a general nature, such as Facebook, Twitter, Instagram, or Snapchat. Again, this suggestion did not seem to be backed by survey results or literature. 

Author’s response: In this part, the authors want to stress the possible setting of human-to-information interaction on the social media site. Please see line 188-196. And especially on line 192, we put some words. "Subsequently, there are options such as whether you wish to set browsing restrictions or are willing to submit to Twitter". And as the reviewer describe that the most popular sites include Facebook, Twitter, Instagram, or Snapchat. So if the Pixiv users can submit their works to Twitter also why it cannot synchronize to another them related social media sites as possible?

Please see Line 586-592.

Comment 5: Overall, the conclusion can be improved by more explicitly drawing links to the research that was completed as part of this project. This would also be a good place to explicitly link to concepts of sustainability, to make clear the relevance of this particular article to the journal.

Author’s response: Yes, thanks for the reviewer’s reminder. This ACG topic research is related to the issues of globalization and cultural diversity. Hope the exploration of the extension of TAM model by Perceived Interactivity to Understand Usage Behaviours on ACG Social Media Sites can help the ACG culture/industry has the scientific and integrated approaches to sustainable development in the future.

Please see line 624-626.
